# Exploiting Enzyme in the Polymer Synthesis for a Remarkable Increase in Thermal Conductivity

**DOI:** 10.3390/ijms24087606

**Published:** 2023-04-20

**Authors:** Anca Petran, Teodora Radu, Monica Dan, Alexandrina Nan

**Affiliations:** Department of Physics Nanostructured Systems, National Institute for Research and Development of Isotopic and Molecular Technologies, 400293 Cluj-Napoca, Romania; anca.petran@itim-cj.ro (A.P.); teodora.radu@itim-cj.ro (T.R.); monica.dan@itim-cj.ro (M.D.)

**Keywords:** poly(benzofuran-co-arylacetic acid), poly(tartronic-co-glycolic acid), crystallinity, Novozyme-435, thermal conductivity, enzymatic reaction

## Abstract

The interest in polymers with high thermal conductivity increased much because of their inherent properties such as low density, low cost, flexibility, and good chemical resistance. However, it is challenging to engineer plastics with good heat transfer characteristics, processability, and required strength. Improving the degree of the chain alignment and forming a continuous thermal conduction network is expected to enhance thermal conductivity. This research aimed to develop polymers with a high thermal conductivity that can be interesting for several applications. Two polymers, namely poly(benzofuran-co-arylacetic acid) and poly(tartronic-*co*-glycolic acid), with high thermal conductivity containing microscopically ordered structures were prepared by performing enzyme-catalyzed (Novozyme-435) polymerization of the corresponding α-hydroxy acids 4-hydroxymandelic acid and tartronic acid, respectively. A comparison between the polymer’s structure and heat transfer obtained by mere thermal polymerization before and enzyme-catalyzed polymerization will now be discussed, revealing a dramatic increase in thermal conductivity in the latter case. The polymer structures were investigated by FTIR spectroscopy, nuclear magnetic resonance (NMR) spectroscopy in liquid- and solid-state (*ss*-NMR), and powder X-ray diffraction. The thermal conductivity and diffusivity were measured using the transient plane source technique.

## 1. Introduction

Polymers are widely used in several areas because of their low weight, low production cost, excellent chemical stability, and, often, inherent functionality. However, their low thermal conductivity (*k*) can forbid special applications such as heat exchangers and electronic packaging [1,2,3,4,5,6,7]. In recent years, the demand for materials with a high thermal conductivity that can dissipate waste heat generated by electronic devices during operation has grown sharply. Most methods for improving the thermal conductivity of polymers have generally focused on composite materials, where additives, such as metal nanoparticles or carbon nanotubes, are incorporated into polymer matrices. However, the increase in thermal conductivity in such composites is usually limited due to the high thermal resistance at the interface between the additives and the polymer matrix. More than 40 years ago, Choy and co-workers demonstrated that the alignment of polymer crystallites could significantly enhance both the mechanical strength and *k* of polymers along the direction of the covalently bonded molecular chains [8,9,10]. Based on previous results, it is expected that molecular alignment produced by gel spinning, drawing, or extrusion used in producing high-performance fibers could significantly enhance *k* [8,9,10,11].

Enzymatic polymerization would be another way to obtain polymers of an increased degree of alignment and, thus, eventually, achieve better thermal conductivity. Besides the fact that polymers with an ordered structure are obtained in enzymatic catalysis, there are other advantages, such as mild reaction conditions, high control of enantio-, chemo-, and regio-selectivity, avoiding undesired side reactions, fewer by-products, and superior catalytic activity toward macrocyclic lactones that are hard to polymerize via chemical catalysis [12,13,14]. Moreover, enzymatic polymerization could help to avoid problems associated with trace residues of metallic catalysts, particularly their unfavorable effects on the environment and toxicity in biomedical applications. Thus, enzymatic polymerization has been regarded as a good platform for green polymer synthesis. Especially the lipase-catalyzed synthesis of aliphatic polyesters has rapidly developed and become an essential synthetic technique [15,16,17] for ring-opening polymerization of lactones and polycondensation, including polycondensation of diacids and hydroxyl acids [18,19,20,21,22,23,24,25]. Undoubtedly, the most important and widely employed biocatalyst for esterification and transesterification in polyester synthesis is lipase B from Candida Antarctica (CALB) in its free and immobilized forms. This was demonstrated in many studies from the 1990s until today. The commercially available Novozyme-435, which exhibits exceptional activity and stability in hydrophobic organic media, is the most commonly used immobilized CALB catalyst [26,27,28]. In this context, the polymers directly obtained by the enzymatic polymerization method will present advantages due to improved thermal properties by increasing the degree of alignment of the polymer chains and reducing production costs.

Recently, poly(benzofuran-*co*-arylacetic acid) (PBAAA) and poly(tartronic-*co*-glycolic) acid (PTGA) was developed by straight forward thermal condensation of 4-hydroxymandelic acid or tartaric acid, respectively. The starting materials are natural products and thus can be obtained from sustainable resources. The synthesis did not request solvents and did not produce harmful side products, just water, i.e., it meets the criteria of green chemistry. So far, these new polymers have not been investigated much and thus await further exploration, with the possibility of practical application. The research presented here was designed to study the difference between polymers synthesized under thermal polycondensation conditions thus far and those synthesized under enzymatic conditions using Novozyme-435. Thermal conductivity (κ) and thermal diffusivity (α), as important parameters related to the ability of the material to transport heat, will be discussed for the two polymers synthesized by the two different methods. In general, the thermal properties of amorphous polymers depend on many factors, such as chemical constituents, structure, type, and strength of defects or structure faults, the strength of bonding, molecular density distribution, the molecular weight of side groups, processing conditions, and temperature. The structure of the polymer synthesized under enzymatic conditions versus the polymer synthesized by thermal polymerization will be discussed based on the FTIR spectra, ss-NMR, and thermal conductivity.

## 2. Results and Discussion

### 2.1. Enzymatic Synthesis of Polymers

Nowadays, the search for new polymers that can find applications in current technology and replace other polymers or be used together with other polymers is a big challenge. poly(benzofuran-*co*-arylacetic acid) **PBAAA** and poly(tartronic-*co*-glycolic) acid **PTGA**, whose properties are not known at all, are two promising candidates in this field.

The thermal polycondensation synthesis of poly(benzofuran-*co*-aryl acetic acid) (**PBAA_Ter**) [29] and poly(tartronic-*co*-glycolic) acid (**PTGA_Ter**) [30] are already described in the literature. The direct enzymatic synthesis of polymers starting from α-hydroxy acids is an attractive alternative as it provides inherent biodegradability and mild reaction conditions for polymerization. It was performed using the same parameters for both monomers (Figure 1). The polymers synthesized under enzymatic conditions were compared with the corresponding polymers obtained by thermal polycondensation both from the structural point of view and the thermal conductivity. In the enzymatic polymerization, Novozyme-435 was employed as a biocatalyst. Novozyme-435 has several unique properties that make it a versatile biocatalyst, perhaps due to the presence of some acid groups in the matrix. Novozyme-435 maintained its catalytic activity and displayed excellent thermal stability up to 100 °C in diphenylether when incubated in toluene at 80 °C for up to a month [31].

As seen from Figure 1, the polymers synthesized under enzymatic conditions consist of the same component units as those synthesized by thermal polycondensation. **PBAAA_Enz** consists of aryl acetic acid and lactone units formed by intramolecular ester formation. However, **PBAAA_Enz** contains a lower number m of lactone rings and, thus, a higher number of aryl acetic units n as the material **PBAAA_Ter** obtained by thermal condensation. This fact is proved by FTIR spectroscopy (see below).

Tartronic acid (**TA**) suffers partial decarboxylation resulting in copolymers **PTGA_Enz** of **TA** and glycolic acid. Here, the ratios between the monomeric units are determined by ^1^H-NMR spectroscopy and found different in both cases (see below).

Regarding PTGA, enzymatic catalysis produces **PTGA_Enz** in a solid form, whereas thermal polycondensation produces PTGA in a honey state. Homo poly(tartronic acid) was previously described in the literature as a solid polymer synthesized by polycondensation of the anhydrosulfite or anhydrocarboxylate of TA obtained with thionyl chloride or phosgene, respectively [32]. Later on, the synthesis was described as accomplished through anionic polymerization of the disodium salt of ketomalonic acid (mesoxalic acid), which is produced by catalytic glycerol oxidation via intermediary TA [33]. Compared with our synthesis, these methods result in different structures and need an additional reaction step with hazardous thionyl chloride or phosgene in the former case.

### 2.2. NMR Spectroscopy of Polymers

The chemical shifts and the shape of the signals in the ^13^C-NMR spectra of **PBAAA_Ter** and **PBAAA_Enz** are almost the same (Figure 1). The differences observed in both spectra are the shape of the signals assigned to the carbon atoms involved in the lactone ring or its neighborhood. One crucial observation is that the carbon signals in the **PBAAA_Enz** spectrum are slightly less broad than in the **PBAAA_Ter** spectrum, meaning there are some small differences in the crystallinity degree of the polymer.

The ^1^H and ^13^C-NMR spectra of **PBAAA_Enz**, recorded in solution, are presented in Appendix A and Appendix A, respectively. Unfortunately, the ^1^H-NMR spectrum was not resolved enough to determine the oligomer ratio.

Comparing the ss-NMR spectra of **PTGA_Enz** with **PTGA_Ter** was impossible because the latter is not solid but in a viscous state. Therefore, the ^13^C ss-NMR spectrum of **PTGA_Enz** was compared with its monomer (TA) (Figure 2). The signals attributed to carbon atoms from the methylene and methine group of the polymer chain are shifted from 71 ppm for the monomer to higher frequencies at 74–77 ppm for **PTGA_Enz**. The ^13^C ss-NMR peaks corresponding to the carbonyl and carboxyl carbon atoms are shifted from 181–180 ppm in **TA** to 169–167 ppm in the polymer **PTGA_Enz**. The signals of the polymers are very narrow, proving that the structure of the polymer is well-ordered.

The ratio between the glycolic unit and tartronic unit present in both **PTGA** polymers was determined from the ^1^H-NMR spectra of **PTGA_Ter** and **PTGA_Enz** (Appendix A). Integration of the methine signal (tartronic acid units) and the methylene signal (glycolic acid units) provided a tartronic acid/glycolic acid ratio of 10/1 for **PTGA_Ter** (Appendix A) and 20:1 in for **PTGA_Enz** (Appendix A), concluding that less decarboxylation had occurred in the enzymatic polymerization. Carboxylic groups are known to form strong hydrogen bonds that enhance intermolecular aggregation and increase melting points. Additionally, reduced decarboxylation in the enzymatic synthesis is likely responsible for the appearance of **PTGA_Ter** as a liquid while **PTGA-Enz** is solid.

Detecting peaks for quaternary carbons lacking directly attached protons is the APT’s primary benefit. Therefore, the assignments of the carbon atoms from **PTGA_Enz** were verified by ^13^C-NMR-APT experiments (Appendix A). As a result, the peaks from the glycolic unit’s methylene (-CH_2_-) carbon atoms group appear orientated up at 59 ppm, whereas the peaks from the tartronic unit’s methine (-CH-) carbon atoms group are oriented down and located at 71.6–73.3 ppm chemical shift. The carbon-atom-specific signals from the carboxyl and ester groups are detected in the 171–175 ppm range.

### 2.3. FTIR Spectroscopy of the Polymers

Figure 3 and Figure 4 compare the FTIR spectra of the thermally synthesized polymers and the enzymatically synthesized ones. Given FTIR spectra were recorded using the same amount of sample, the intensity of the adsorption bands can be used as a quantitative measure. In the case of **PBAAA_Enz**, (Figure 3), it is observed that the absorption band at 1800 cm^−1^ attributed to the C=O bond in the lactone ring is less intense than that of the polymer **PBAAA_Ter** synthesized by thermal condensation. By deconvolution of the two adsorption bands (1725 cm^−1^ for **PBAAA_Enz**, 1735 cm^−1^ for **PBAAA_Ter** for COOH, and 1800 cm^−1^ the case of both polymers for lactone moieties) specific to C=O bonds (Appendix A), the percentage contribution of the two bands were determined. It clearly demonstrates a much smaller contribution of the C=O bond belonging to the lactone group in the case of **PBAAA_Enz** synthesized under enzymatic conditions (Table 1).

In the FTIR spectrum of the enzymatically synthesized polymer **PBAAA_Enz**, the intense and wide band specific to C=O bonds at 1730 cm^−1^ is pronounced. The ester group absorption bands (-COO-) are present in both FTIR spectra of **PBAAA** polymers at the same wavenumber, 1614 cm^−1^. Other important adsorption bands present in both FTIR spectra are at 1516 cm^−1^ and 1489 cm^−1^ and ascribe the C–C stretch (in-ring) of the benzene ring.

In the FTIR spectra of **PTGA_Enz** and **PTGA_Ter**, are some slight differences between the adsorption bands. The most important adsorption band is at 1738 cm^−1^, specific for the -C=O stretch from the carboxyl and ester group. In the **PTGA_Enz** FTIR spectrum, we can observe a wider and more intensive adsorption band at 3400 cm^−1^ specific for the associate -COO-H group than in the **PTGA_Ter** FTIR spectrum, suggesting that the **PTGA_Enz** polymer structures include more carboxyl groups. The -C-O stretching modes in the ester group appear at 1219 cm^−1^, while the C-O-C asymmetric mode appears at 1118 cm^−1^. The shoulder at the wavelength of 1628 cm^−1^ that appears more visibly in the FTIR spectrum of **PTGA_Enz** is attributed to the ester groups present in the polymer chain.

### 2.4. Powder XRD Analysis

X-ray diffraction (XRD) analysis on samples of polymers provides important structural information, such as the degree of crystallinity. To determine the polymer crystallinity, we use an empirical two-phase model that considers the polymer under study as made entirely of crystalline and amorphous phases, with no regions of semicrystalline structures. The degree of crystallinity of polymer-based materials is determined using the formula:χ_c_ = [I_c_/(I_a_ + I_c_)] × 100%,(1)
where I_c_ and I_a_ are diffraction intensities of crystalline and amorphous phases, respectively.

The diffraction patterns of **PBAAA_Enz**, shown in Figure 5, display a broad maximum around 2θ = 20°, indicative of a dominant amorphous structure with small crystalline peaks at 19.5, 20.9, and 21.8°. In the case of **PBAAA_Ter**, the XRD curve displays a broad maximum of around 21.2°, indicative of an amorphous structure and a sharp crystalline peak at 17.4°. Using Formula (1), the degree of crystallinity was determined, X_c_ = 29% for **PBAAA_Enz** and X_c_ = 19% for **PBAAA_Ter**, respectively, indicating that the crystallinity is higher in the polymer synthesized in the enzymatic condition where more crystalline peaks are present. The *ss*-NMR spectroscopy was also used to demonstrate no significant changes in crystallinity structure between the two **PBAAA** polymers.

Figure 6 compares the powder XRD spectra of tartronic acid (**TA**) and **PTGA_Enz**, illustrating the polymer’s crystalline nature. We decided to compare the XRD spectra of the polymer and monomer to clearly demonstrate that even though the polymer **PTGA_Enz** has very well-defined crystalline peaks, they do not overlap those of the monomer. The fact that the peak position does not coincide shows that the monomer and polymer have different structures; in other words, the polymerization process took place in the entire volume of the material. The data obtained by powder XRD are in good agreement with those obtained by the ss-NMR spectroscopy.

### 2.5. Thermogravimetric Analyses of Polymers

Figure 7 shows the TGA curves of the enzymatically and thermally synthesized **PBAAA**. They were obtained in the air from room temperature to 800 °C. Regarding thermal stability, the thermally synthesized **PBAAA_Ter** is a bit more stable than the one obtained in the enzymatic condition. Under conditions of thermal stress, the **PBAAA_Enz** presents 4 stages: a mass loss of 12 % in the temperature range of 40–160 °C, corresponding to the elimination of water molecules absorbed in the intra-lamellar space, followed by a second loss of 14% in the range 160–280 °C, corresponding to massive decarboxylation reactions in the polymer structure. The third stage, with a mass loss of 36 % in the range 280–500 °C, is characteristic of the rupture of the polymeric chains, and the last 38% stage in 500–720 °C is associated with the total decomposition of polymeric chains. The weaker thermal stability of **PBAAA_Enz** may come from the fact that the number of lactone rings m formed by the elimination of intermolecular water from the polymer chain is much smaller than in the case of **PBAAA_Ter**.

Appendix A presents the DTA curves of **PBAAA_Ter** and **PBAAA_Enz**, showing the decomposition process of both polymers. The **PBAAA_Ter** decomposition occurs in the temperature range of 245–575 °C with a very exothermic peak in the DTA, while the decomposition of **PBAAA_Enz** starts at 442 °C and finishes at 600 °C with 3 intense exothermic peaks.

The thermogravimetric analysis of **PTGA_Enz** shows a thermally unstable polymer (Appendix A), which undergoes fast degradation starting from 160 °C and loses 90% of the total mass until around 300 °C. This low thermal stability is due to the presence of a larger number of tartronic units in the final polymer chain compared to **PTGA_Ter**, which could not be investigated because of its hone-like consistency. The presence of a large number of carboxyl groups in **PTGA_Enz** leads to the massive elimination of carbon dioxide during the thermogravimetric analysis starting at 200 degrees Celsius.

The DTA (Figure 8) of **PTGA_Enz** reveals a melting point at T_m_ 165 °C, not found in the starting material. The shoulder at around 80 °C could be interpreted as a glass transition T_g_. With this assumption, the ratio of T_g_/T_m_ would be 80. This is higher than common polymers but not unusual [34]. The maxima at higher temperatures are certainly caused by decomposition reactions.

### 2.6. Thermal Analysis of Polymers

The TPS method was used to analyze the thermal properties of the obtained samples at room temperature (RT), 50 °C, and 100 °C. The effect of temperature change on the thermal properties of studied polymeric materials is complex. In general, many efforts were made to obtain polymers’ high thermal conductivity, but almost all with polyethylene fibers. In the laboratory, polyethylene with high κ values (9–104 W/(mK)) has already been attained [10,11,35,36,37]. This depends on several factors, such as synthesis method, structure, degree of crystallinity, etc. **PBAAA_Ter** has lower thermal conductivity κ and diffusivity than **PBAAA_Enz** at each investigated temperature (Figure 9). This may be related to a reduced degree of crystallinity observed for the **PBAAA_Ter**, considering that defects and disordered chains in the polymer structure reduce their thermal conductivity by hampering the phonon transport along the polymer chains. It can be observed that for **PBAAA_Ter**, the temperature variation does not influence the conductivity values κ. They remain almost constant at around 0.17 W/(mK). **PBAAA_Enz**, on the other hand, suffers from changes in thermal conductivity values depending on temperature. At room temperature, it is 55% higher than for **PBAAA_Ter**. The increase of κ from 0.264 to 0.341 at 50 °C can be explained by a directly proportional increase of the crystalline fraction present in **PBAAA_Enz**, which improves phonon transfer along the chains. The thermal diffusivity at RT increases from 0.123 mm^2^/s for **PBAAA_Ter** to 0.411 mm^2^/s in the case of **PBAAA_Enz**, which means a 70% increase in heat rate transfer of the polymer from the hot end to the cold end. The fact that the thermal diffusivity also increases in direct proportion to the thermal conductivity confirms that the **PBAAA_Enz** polymer undergoes an improvement in structure by creating crystalline structural entities. At 100 °C, the previously aligned chains break into a more disordered state, resulting in a decrease of κ to 0.186.

The most impressive results were obtained in the case of **PTGA_Enz**, where, at room temperature, a more than 3 times higher thermal conductivity was observed than for **PTGA_Ter** of κ (κ = 1.048 W/(mK versus 0.287 W/(mK) (Figure 10). A value of = 1.048 W/(mK) for a synthetic polymer is extremely high, given that ordinary synthetic polymers usually fall in the range of 0.1–0.3 W/(mK) [38,39,40,41] as a result of the complicated architecture of polymer chains. Because the experiment was a 1-sided test and high temperatures were not permitted, the κ value could not be measured for **PTGA** polymers at 100 °C. Nevertheless, the κ value at RT and 50 °C are almost equal. Therefore, we could conclude that the temperature increase does not affect the κ values of the crystallin polymer **PTGA_Enz**, which means that the phonon transfer along the polymer chains does not occur in this case. In the instance of **PTGA_Enz** compared to **PTGA_Ter**, the thermal diffusivity rises by 70%, demonstrating a potent heat rate transmission of the crystalline structure created in the **PTGA_Enz** polymer structure.

## 3. Materials and Methods

### 3.1. Chemicals and Reagents

The monomers and the biocatalyst lipase immobilized on acrylic resin from *Candida antarctica* (Novozym 435, ≥10,000 U/g) used in this study were purchased from Sigma Aldrich (St. Louis, MO, USA) and Alfa Aesar by ThermoFisher Scientific (Kandel, Germany) and did not require further purification.

### 3.2. Enzymatic Synthesis of Polymers

#### 3.2.1. Enzymatic Synthesis of PBAAA_Enz

*p*-Hydroxymandelic acid monohydrate (HMA) (1 g, 5.37 mmol) and Novozyme-435 (100 mg) were added into a single-neck round-bottom flask in 5 mL distilled water, and the mixture was magnetically stirred for 5 min. Then, toluene (110 mL) was added, and the mixture was left under magnetic stirring for 5 days at 78 °C. The resulting suspension was filtered in order to remove the enzyme from the reaction mixture. The filtrate was evaporated in a rotary evaporator until the total removal of the solvents left back a purple solid precipitate. To remove traces of water or toluene from this precipitate, methanol was added and evaporated once more until the solvent was completely removed, obtaining 0.9 g of a purple precipitate of PBAAA_Enz. ^1^H NMR (500 MHz, (CD_3_)_2_SO, δ, ppm): 4.6–5.6 (m; HO-CH(Ph)-COOH, Ph-CH(COOH)-Ph and -CH_from the lactone ring_); 6.0–7.6 (m, -CH_aromatic_); 9.3–10 (-COOH). ^13^C-NMR (500 MHz, (CD_3_)_2_SO, δ): 49–52 (Ph-CH(COOH)-Ph); 56–57 (-CH_from the lactone ring_); 72–81 (HO-CH(Ph)-COOH); 115–158 (-CH_aromatic_ and -C_aromatic_); 171–174 (-COOH and -COO-).

#### 3.2.2. Enzymatic Synthesis of PTGA_Enz

In a flask, a mixture formed from tartronic acid (TA) (1 g, 8 mmol), Novozyme-435 (100 mg), and 5 mL distilled water was stirred for 5 min in a single-neck round-bottom flask. Toluene (110 mL) was added. After magnetic stirring at 78 °C for 5 days, the enzyme was removed by suspension and filtration. The filtrate evaporated in a rotary evaporator leaving behind a white precipitate. To remove traces of water or toluene from this precipitate, methanol was added and evaporated to complete dryness obtaining **PTGA_Enz** (0.83 g) as a white sticky solid. ^1^H NMR (500 MHz, (CD_3_)_2_SO, δ, ppm): 3.8 (-CH_2_- from the glycolic unit); 4–4.6 (-CH- from tartronic unit). ^13^C-NMR (500 MHz, (CD_3_)_2_SO, δ, ppm): 59 (-CH_2_- from the glycolic unit); 71.6–73.3 (-CH- from tartronic unit); 171–175 (-COOH and -COO-).

### 3.3. NMR Spectroscopy

The liquid nuclear magnetic resonance (NMR), ^1^H, ^13^C-NMR, and attached proton test (APT) NMR spectra were recorded at room temperature in deuterated dimethyl sulfoxide ((CD_3_)_2_SO) as a solvent in a 5 mm tube on a 500 MHz NMR spectrometer Bruker Biospin GmbH.

Solid-state ^13^C NMR (^13^C ss-NMR) spectra were recorded at 125.73 MHz Larmor frequency with a Bruker Advance III 500 MHz wide-bore NMR spectrometer operating at room temperature, using a 4 mm double resonance (1H/X) MAS probe, the sample being packed in 4 mm zirconia rotors. Peak assignments were performed through simulation using ACD/Labs 12.00 software (Product Version 12.5, Build 47877, 20 April 2011). Standard ramped-amplitude cross polarization in magic-angle-spinning (RAMP CP-MAS) NMR spectra were acquired at 14 kHz spinning frequencies, 2 ms contact times, and proton decoupling under a two-pulse phase-modulated (TPPM) sequence.

### 3.4. FTIR Spectroscopy

The FTIR spectra were measured by a JASCO FTIR spectrophotometer model 6100 in the 5000–350 cm^−1^ wavenumber range by incorporating the sample into KBr (IR grade) tablets.

### 3.5. Thermogravimetric Analysis

Thermogravimetry measurements were performed in air, using TA Instruments SDT Q 600 equipment, in the temperature range from 30 to 800 °C with a heating rate of 10 °C min^−1^.

### 3.6. Powder XRD

X-ray patterns were collected at room temperature using a Rigaku SmartLab multipurpose diffractometer with Cu Kα1 radiation (λ= 1.54056 Å), equipped with a 9-kW rotating anode. For the acquisition of the experimental data, SmartLab Guidance software was used. The measurements were performed in the 5–90° range in steps of 0.01°.

### 3.7. Thermal Conductivity and Diffusivity

The thermal conductivity was measured in a Hot Disk TPS 2500S (Hot Disk AB, Kagaku, Sweden) apparatus with a 5464F1 sensor using the transient plane source (TPS) method. The equipment has included determining the diffusivity and specific heat of materials. The method’s principle consists of applying a short heat pulse of a predetermined duration to the sample, initially kept at thermal equilibrium using a TPS sensor with a double function: constant heat source and temperature sensor placed between two identical samples. The transient temperature response of the samples is recorded and further used to estimate the thermal conductivity. To get results with excellent accuracy, the samples were prepared in the form of identical pellets with a radius of 5 cm and a thickness of about 4 mm to ensure that we could use a TPS sensor with a diameter of 2 mm so that the heat generated by the spiral area does not diffuse to the sample outside boundary within a predefined period of measurement time.

## 4. Conclusions

This work investigated the structural and properties differences between poly(benzofuran-*co*-arylacetic acid) **PBAAA** and poly(tartronic-*co*-glycolic acid) **PTGA** previously synthesized from 4-hydroxymandelic acid or tartaric acid, respectively, under thermal polycondensation conditions (**PPAAA_Ter** and **PTGA_Ter)** obtained now under enzymatic conditions using Novozym-435 (**PBAAA_Enz** and **PTGA_Enz**). NMR spectroscopy, FTIR spectroscopy, powder XRD diffraction, and TPS measurements were performed. Polymers of different structures and properties were obtained. The most remarkable results were obtained in the case of **PTGA**, where an increase of thermal conductivity κ at room temperature was from 0.287 W/(mK) for **PTGA_Ter** to 1.048 W/(mK) **PTGA_Enz**, which in percentage is a 265% increase. This presents an outstanding value for a polymer that usually ranges between κ = 0.1–0.3 W/(mK). To understand this κ enhancement, the degree of crystallinity was determined using powder XRD measurements. This allowed conclusions about the influence of polymer chain alignment on thermal transport. The findings in the heat transfer enhancement are important for applications in thermal management, such as polymer heat exchangers, chip packaging, soft robotics, and organic light-emitting diode and catalyst devices.

## Data Availability

All data not present in the main text or the ESI are available from the authors upon request.

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
