# Peer review of "Exploiting Enzyme in the Polymer Synthesis for a Remarkable Increase in Thermal Conductivity"

_ijms, 2023, doi:10.3390/ijms24087606_

Round 1

Reviewer 1 Report

The manuscript entitled: "Exploiting enzyme in the polymer synthesis for a remarkable increase in thermal conductivity" reports on the utilization of Novozim 435 in preparation of two class of polymers: poly(benzofuran-co-arylacetic acid) and poly(tartronic-co-glycolic acid) proving high thermal conductivity. The structural characteristics as well as the thermal conductivity and diffusivity were also compared with the  polymers obtained by melt-polymerization technique.

The experimental design of the polymers and the results are well presented and correlated. I identified only minor issues to be addressed:

-in Introduction section, at line 45 please add a reference to support the idea.

-There are some repetitions in the first paragraph at page 2, lines 46-60, regarding the advantages of enzymatic polymerization, so that I recommend a better reorganization of the content.

-There is no motivation on the choice of the monomers to prepare the polymers. I recommend the author to emphasize better their ideas in the exploitation of these classes polymers.

-in Section 2.1, the reagents used must be mentioned. Some characteristics regarding the stability of Novozim-435 must be added.

-in Section 2.2. the catalyst Novozim is stable at the synthesis temperature? How was inactivated the catalyst? only be filtration?

-Section 3.2 reports both NMR and FTIR results, some corrections are necessary.

-In FTIR section, some peaks are necessary in Figs. 3 and 4.

- In section 3.4 how authors explain the lower thermal stability of the polymer generated by enzymatic polymerization? The composition seems to be the same, as it can be seen in Figs. 3 and 4.

-in Section 3.5 some literature comparisons are necessary.

Based on these observations I recommend the acceptance of the manuscript after Minor revision.

Reviewer 2 Report

Journal Title: International Journal of Molecular Science

Manuscript Title: Exploiting enzyme in the polymer synthesis for a remarkable increase in thermal conductivity

Manuscript ID: ijms-2319303

Authors: Anca Petran et al.

The current paper describes some comparative extension studies on two compounds prepared by some different procedures. The main claimed objective here was to develop polymers with high thermal conductivity.

The authors prepared two polymers by enzyme-assisted polymerization of 4-hydroxymandelic acid and 2-hydroxymalonic acid, respectively, then, they compared some properties of the resulted products with the properties of same products obtained by solution or thermal polycondensation, referring their own former works (Polym. Chem., 2017, 8, 3504–3514 and Polym. Int., 2018, 67, 212–219). Nevertheless, despite their comparison, while reading the paper, the reader remains with the impression that the products are quite different. For example, PTGA_Enz is a solid product while PTGA_ter is a viscous product. Why? Because the products have different structure, or they are a mixture of oligomers. As a consequence, it is not appropriate to compare results in Figure 8 and Figure 9. Moreover, if Figure 8 is merged with Figure 9, there will be evident that the ”remarkable” increase in thermal conductivity in sample PTGA_Enz is actually approximately comparable with k values  of PBAAA_Ter.

Unfortunately, the current version of the manuscript is successfully rejectable.

Here are my additional critical remarks:

-Abstract, page 1, line 10: ”good resistance” what kind of resistance have the authors quantified as good?

-Check the similarity of ”The demand for polymers with high thermal conductivity has grown due to their low density, low cost, flexibility, and good resistance.” with ”Polymers are usually known for their low thermal conductivity. However, the demand in industries for polymers with high thermal conductivity has increasingly grown due to their low density, low cost, flexibility, and good environmental resistance compared with conventional substances of high thermal conductivity.” - Journal of Enhanced Heat Transfer Volume 27, 2020 Issue 5 THERMAL CONDUCTIVITY ENHANCEMENT OF POLYMERS VIA STRUCTURE TAILORING.

-Abstract, page 1, line 11: ” Improving” ”I”  is bold

-”This research aimed to develop polymers with a high thermal conductivity that are interesting for several applications.” In the end only one polymer can be considered for this purpose. Also, some ”several applications” should be mentioned.

-Line 17: ”corresponding α-hydroxy acids 4-hydroxymandelic acid or tartronic acid” should be ” corresponding α-hydroxy acids, 4-hydroxymandelic acid or tartronic acid”.

-Line 18: ”melt-polymerization” is not adequate terminology here as sample PBAAA_Ter was prepared by solution polycondensation (reflux toluene), the authors may refer distinctively between PBAAA_Ter and PTGA_ter which indeed was prepared by melt-polycondensation.

-Introduction: Check the similarity of ”Polymers are widely used in industry and our daily life because of their diverse func- 27 tionality, lightweight, low cost, and excellent chemical stability. However, in some appli- 28 cations such as heat exchangers and electronic packaging [1-7], polymers’ low thermal 29 conductivity (k) is one of the major technological barriers and is one of the most important 30 parameters of polymeric materials.” It seems more or less copy-paste from https://doi.org/10.1016/j.mser.2018.06.002

-The authors have the obligation to check the similarity of the text along the entire content of the manuscript!

-line 37: check the meaning of ” More than 40 years ago, already”

-the sentences 42-45 need reference(s)

-line 67: ” This study was designed to study” please fix

-In my opinion, ”the difference between polymers previously synthesized under thermal polycondensation conditions and those synthesized under enzymatic conditions using Novozym-435” is that they are different!

-line 88: ”p-Hydroxymandelic acid (HMA) (1g, 5.37 mmol)”. The molar mass of HMA = 168.15… 1g is 5.94 mmol

-line 89, what kind of flask? What kind of water?

-line 88-93, HMA and Novozyme-435 were dissolved in water. Later, the authors said ” The resulting suspension was filtered in order to remove the enzyme from the reaction mixture.” Was it dissolved was it not? Please be generous when describing the synthetic pathway.

-line 100, 2.2.2. Enzymatic synthesis of PTGA_Enz: there is a question raised when comparing the synthesis of the compound by the two methods. The product made by melt-polymerization was purified, as published in Polym. Int., 2018, 67, 212–219, by methanol precipitation to eliminate the poly(glycolic acid) by-product (white solid), while the filtrate was concentrated to give the oil sticky PTGA_ter compound. Now, by enzyme-assisted polymerization the authors took the white precipitate resulted after the filtration of enzyme from suspension and washed it with methanol but this time they worked up the white solid. Which is the probability to be a mixture of PTGA_Enz with  poly(glycolic acid) or plenty of oligomers? I would like to see and compare some NMRs on the two ”white powders”.

-page 4, 3.1. Enzymatic synthesis of polymers. ”The thermal polycondensation synthesis of poly(benzofuran-co-aryl acetic acid) (PBAA_Ter)” is indeed presented in literature but there where different conditions with different synthetic pathways. Now the authors come with a pure and plain structure for PBAAA_Enz and PTGA_Enz.

-line 170-178: the paragraph raises another tricky question: was the yellow sticky oil reported previously by the authors indeed the poly(tartronic-co-glycolic acid) product if it must be a solid as other referred studies reported

-how was the time of reaction set at 5 days? Why 5 days?

-the structure of PBAAA_Enz as presented in Scheme 1 require more in-depth studies.

-it is recommended to perform DSC measurements as the possible transitions will enlighten the amorphous/crystalline character. Same, the thermal transition could be correlated with the melting point of the starting ingredient.

-GPC measurements are also of interest, to check the distribution of molecular weights as the hyper-ramification behavior is very likely to interfere in this case

-line 242, the afirmation ”the PTGA_Enz polymer structures include more carboxyl groups.” needs reference

- 3.5. Thermal analysis of polymers. The temperature differenced studies on thermal conductivity and thermal diffusivity strictly impose differential scanning calorimetry studies as the afirmation ”The fact that the thermal diffusivity also increases in direct proportion to the thermal conductivity confirms that the PBAAA_Enz polymer undergoes an improvement in structure by creating crystalline structural entities. At 100 ⁰C, the previously aligned chains break into a more disordered state resulting in a decrease of κ to 0.186.”

-line 359 ” and organic light-emitting diodes and catalyst devices.” Too many and…

Round 2

Reviewer 2 Report

The authors have responded to most of the issues raised and their manuscript has increased in quality and scientific soundness, being now acceptable to be published in IJMS.